# Routers in Vision Mixture of Experts: An Empirical Study

**Tianlin Liu**[*] *University of Basel*

**Mathieu Blondel** *Google DeepMind*

**Carlos Riquelme**[†] *Stability AI*

**Joan Puigcerver** *Google DeepMind*

[*]*Work done as an intern at Google DeepMind.* [†]*Work done while at Google DeepMind.*

**Reviewed on OpenReview:** *https://openreview.net/forum?id=aHk3uctnf1&noteId=vSY7Pa3Bbf*

## Abstract

Mixture-of-Experts (MoE) models are a promising way to scale up model capacity without significantly increasing computational cost. A key component of MoEs is the **router**, which decides which subset of parameters (experts) process which feature embeddings (tokens). In this paper, we present a comprehensive study of routers in MoEs for computer vision tasks. We introduce a unified MoE formulation that subsumes different MoEs with two parametric **routing tensors**. This formulation covers both **sparse MoE**, which uses a binary or hard assignment between experts and tokens, and **soft MoE**, which uses a soft assignment between experts and weighted combinations of tokens. Routers for sparse MoEs can be further grouped into two variants: Token Choice, which matches experts to each token, and Expert Choice, which matches tokens to each expert. We conduct head-to-head experiments with 6 different routers, including existing routers from prior work and new ones we introduce. We show that (i) many routers originally developed for language modeling can be adapted to perform strongly in vision tasks, (ii) in sparse MoE, Expert Choice routers generally outperform Token Choice routers, and (iii) soft MoEs generally outperform sparse MoEs with a fixed compute budget. These results provide new insights regarding the crucial role of routers in vision MoE models.

Deep learning is using ever larger models. However, larger models require extensive computational resources for training and deployment. To further scale up model capacity without linearly increasing the computational cost, one promising direction is to use mixture-of-experts (MoE) layers in neural networks. The general idea of a MoE layer is to assign different feature embeddings, also known as **tokens**, to subsets of neural network parameters, known as **experts**. Compared to traditional dense models, MoEs yield better performance–compute trade-off (Shazeer et al., 2017; Riquelme et al., 2021; You et al., 2021; Mustafa et al., 2022; Puigcerver et al., 2023).

At the heart of the MoE model is its **router**, responsible for matching tokens with experts. Routers used before the deep learning era were reviewed in Yuksel et al. (2012), where they termed a router as a gate. In the deep learning era, Bengio et al. (2016) cast the input-to-expert routing problem as a Markov decision process, and used reinforcement learning to train the router. Shazeer et al. (2017); Lepikhin et al. (2021); Fedus et al. (2022b) introduce differentiable training approaches of routers without the complication of reinforcement learning. Roller et al. (2021) use a non-learnable router that hashes input samples in a deterministic way. Clark et al. (2022); Kool et al. (2021); Liu et al. (2023) parametrize routers based on optimal transport formulations. Sander et al. (2023) proposed differentiable top-$k$ operators that help the learning of token–expert matching. All aforementioned examples are **sparse MoEs** that seek a hard, binary token–expert match. In contrast, the recently proposed **soft MoE** (Puigcerver et al., 2023) router matches linearly

combined inputs with experts. This soft MoE approach avoids the discrete, non-differentiable transforms often used in sparse MoEs while having a similar computational cost.

MoEs as layers in deep neural networks were first explored in language modeling (Shazeer et al., 2017) before being applied to computer vision tasks (Riquelme et al., 2021). It is natural to hypothesize that many language MoE routers can be adapted to work well in vision MoE models, given that both rely on transformer architectures (Vaswani et al., 2017; Dosovitskiy et al., 2021). However, we currently lack a comprehensive comparison between routers specifically designed for vision MoEs and those repurposed from language MoEs. For instance, the Sinkhorn router (Clark et al., 2022), initially introduced for language MoE, can be potentially used for vision tasks as well; but to the best of our knowledge, their performances have not been reported in prior vision MoE work. In this work, we address this gap by conducting an empirical study of routers in vision MoEs. We compare different vision MoE models head-to-head by varying their underlying routers. We then evaluate these models through few-shot transfer learning and fine-tuning on ImageNet, providing a comprehensive analysis of the performance of different routers in image recognition tasks. The main contributions of this paper are summarized below.

- We propose a unified formulation of MoE layers by way of parametrizing its **dispatch tensor** and **combine tensor**, which we collectively refer to as **routing tensors**. This generalizes the dispatch weights and combine weights of Puigcerver et al. (2023), which focuses on Soft MoE. This unified approach allows for a comparison of existing MoE routers and motivates us to introduce new routers.

- We study sparse MoEs by decoupling **(i)** how they parameterize token-expert affinity matrices (e.g., based on softmax and Sinkhorn transform) and **(ii)** how they determine which experts to use: Expert Choice (where each expert selects tokens) and Token Choice (where each token selects experts). We demonstrate that factor **(ii)** outweighs factor **(i)** in sparse MoE routers.

- We show that the soft MoE router generally outperforms the sparse variants—both previously reported and our newly introduced one in Section 2.4—under a fixed compute budget.

**Notations.** Throughout this paper, we use bold letters for vectors, matrices, and tensors. For a vector $\boldsymbol{a}$, we let $\boldsymbol{a}[i]$ be its $i$-th entry. For a matrix $\boldsymbol{A} \in \mathbb{R}^{m \times n}$, we let $\boldsymbol{A}[i, j]$ be its $(i, j)$-th entry, let $\boldsymbol{A}[:, j] \in \mathbb{R}^m$ be its $j$-th column, and $\boldsymbol{A}[i, :] \in \mathbb{R}^n$ be the $i$-th row of $\boldsymbol{A}$, transposed into a column vector. We write $[N] \coloneqq \{1, 2, \ldots, N\}$. We denote an all-one vector in $\mathbb{R}^D$ by $\mathbf{1}_D$.

The paper is structured as follows: Section 1 details a unified formulation of Mixture of Experts (MoE) layers, defined through their routers. Section 2 explores various instantiations of MoE layers within this formulation. Section 3 presents the numerical experiments conducted on MoE layers.

# 1 A Unified Formulation of MoE Layers

In this section, we first introduce the most commonly used router, which is based on the composition of the softmax function and the top-k function. We then provide a unified formulation of MoE layers that encompasses many types of MoE layers as special cases. Throughout our work, without loss of generality, we let each expert be a Multi-Layer Perceptron (MLP).

## 1.1 A motivating example of MoE layer

In the most widely used MoE formulation, each token $\boldsymbol{x} \in \mathbb{R}^D$ chooses $k$ experts through a softmax function (Shazeer et al., 2017; Fedus et al., 2022b; Riquelme et al., 2021; Allingham et al., 2022):

$$\mathsf{MoE}(\boldsymbol{x}) \coloneqq \sum_{r=1}^{E} \mathsf{Gate}_r(\boldsymbol{x}) \cdot \mathsf{MLP}_r(\boldsymbol{x}) \quad \text{with} \quad \mathsf{Gate}_r(\boldsymbol{x}) \coloneqq \mathsf{top}_k\Big(\mathsf{softmax}(\boldsymbol{W}\boldsymbol{x} + \boldsymbol{\epsilon})\Big)[r] \in \mathbb{R}, \ \forall r \in [E], \quad (1)$$

where the gating weights $\left\{\mathsf{Gate}_r(\boldsymbol{x})\right\}_{r=1}^{E}$ linearly combine the outputs of the $E$ experts $\{\mathsf{MLP}_r(\boldsymbol{x})\}_{r=1}^{E}$. The function $\mathsf{top}_k : \mathbb{R}^E \to \mathbb{R}^E$ sets all but the $k \ll E$ largest numbers to zero, thereby selecting few experts for the

combination. The gate parameters $\boldsymbol{W} \in \mathbb{R}^{E \times D}$ are trained in conjunction with other network parameters. The vector $\boldsymbol{\epsilon} \sim \mathcal{N}(0, \sigma^2 \boldsymbol{I})$ is a noise injected to the logits $\boldsymbol{XW}$, where $\sigma \in \mathbb{R}$ controls the strength of the noise; through auxiliary losses, the noise helps to improve the numerical stability of router (Appendix A). We refer to the formulation (1) as **Softmax Token Choice** MoE (not be confused with Soft-MoE, described in Section 2.6). This name emphasizes that each token selects $k$ experts based on softmax scores.

Compared to a traditional multi-layer perceptron (MLP) layer, the MoE layer (1) offers a more flexible balance between the model's ability to fit the data and computational cost. Evaluating the MoE layer is equivalent to evaluating a fixed number of MLP layers in parallel, where the number of layers is determined by the value of $k$. By keeping $k$ fixed, as the number of experts $E$ increases, the MoE layer's ability to fit the data increases, but the computational cost remains roughly the same, up to the computation of the gating weights $\mathsf{Gate}_r(\boldsymbol{x})$. In this way, the MoE layer provides a way to control the model-fitting capacity and computational cost trade-off by adjusting the number of experts.

## 1.2 Unifying MoE layers through routers

We now introduce a unified formulation of MoE layers, which encompasses the Softmax Token Choice layer and other MoE layers as special cases. This formulation is for a minibatch setting, where a MoE layer processes a group of $T$ tokens $\{\boldsymbol{x}_1, \ldots, \boldsymbol{x}_T\} \subset \mathbb{R}^D$ in parallel. An additional hyperparameter used in this minibatch setting is the **buffer capacity** of experts. It specifies the maximum number of tokens that each expert can handle in a minibatch. For efficient hardware usage, it is ideal for each expert to have a small buffer capacity $C \ll T$, such as $C = \lceil T/E \rceil$ or $C = \lceil 2T/E \rceil$ (Riquelme et al., 2021). A minibatched MoE layer with a buffer capacity $C$ is defined as follows.

---

**A Unified Formulation of MoE Layers**

Let $\{\boldsymbol{x}_1, \ldots, \boldsymbol{x}_T\} \subset \mathbb{R}^D$ be a minibatch of $T$ tokens and let $\boldsymbol{X} \in \mathbb{R}^{T \times D}$ be a matrix of row-wise concatenation of tokens with $\boldsymbol{X}[t, :] = \boldsymbol{x}_t$. Let $\boldsymbol{D_X} \in \mathbb{R}^{T \times E \times C}$ and $\boldsymbol{C_X} \in \mathbb{R}^{T \times E \times C}$ be certain **dispatch** and **combine** tensors that are functions of tokens $\boldsymbol{X}$. A MoE layer with a fixed buffer capacity $C \in \mathbb{N}$, dispatch tensor $\boldsymbol{D_X}$ and combine tensor $\boldsymbol{C_X}$ is defined as

$$\mathsf{MoE}(\boldsymbol{X})[t, :] := \sum_{r=1}^{E} \sum_{c=1}^{C} \boldsymbol{C_X}[t, r, c] \, \mathsf{MLP}_r\left(\boldsymbol{X}^\top \boldsymbol{D_X}[:, r, c]\right) \in \mathbb{R}^D. \tag{2}$$

---

Here, $\boldsymbol{D_X}$ is termed dispatch tensor, since it is responsible for sending different tokens to different experts; $\boldsymbol{C_X}$ is called the combine tensor, as it is used to linearly combine the expert outputs.

**Recovering the Softmax Token Choice layer as a special case.** On a single token level, the Softmax Token Choice layer in (1), can be recovered using the more general formulation (2) by the following combine tensor $\boldsymbol{C_X}$ and the dispatch tensor $\boldsymbol{D_X}$:

$$\boldsymbol{C_X}[t, r, c] := \begin{cases} \mathsf{Gate}_r(\boldsymbol{x}_t), & \text{if } t = c, \\ 0, & \text{otherwise,} \end{cases} \quad \text{and} \quad \boldsymbol{D_X}[t, r, c] := 1\left(\boldsymbol{C_X}[t, r, c] > 0\right). \tag{3}$$

With these choices, the MoE output in (2) is then

$$\begin{aligned} \mathsf{MoE}(\boldsymbol{X})[t, :] &= \sum_{r=1}^{E} \sum_{c=1}^{C} \boldsymbol{C_X}[t, r, c] \cdot \mathsf{MLP}_r\left(\boldsymbol{X}^\top \boldsymbol{D_X}[:, r, c]\right) \\ &= \sum_{r=1}^{E} \boldsymbol{C_X}[t, r, t] \cdot \mathsf{MLP}_r\left(\boldsymbol{X}^\top \boldsymbol{D_X}[:, r, t]\right) \\ &= \sum_{r=1}^{E} \mathsf{Gate}_r(\boldsymbol{x}_t) \cdot \mathsf{MLP}_r\left(\boldsymbol{x}_t\right), \end{aligned} \tag{4}$$

which recovers the Softmax Token Choice layer (1) on a single-token level. Note that in (3) we use $C = T$. In practical implementation, however, we usually choose $C \ll T$ to reduce the computational cost. This is detailed in Section 2.1.

**MoE routers.** The Softmax Token Choice router uses a specific combine tensor $\boldsymbol{C_X}$ and the dispatch tensor $\boldsymbol{D_X}$. But they can be parametrized in more flexible ways as generic transforms of $\boldsymbol{X}$. We refer to this mapping $\boldsymbol{X} \mapsto (\boldsymbol{C_X}, \boldsymbol{D_X})$ as a **MoE router**, which is central to our study. It is defined as follows.

---

**MoE Router**

A **MoE router** takes minibatch input $\boldsymbol{X} \in \mathbb{R}^{T \times D}$ and yields the dispatch tensor $\boldsymbol{D_X} \in \mathbb{R}^{T \times E \times C}$ and the combine tensor $\boldsymbol{C_X} \in \mathbb{R}^{T \times E \times C}$ that are used by the MoE layer (2):

$$\text{Router} : \boldsymbol{X} \mapsto (\boldsymbol{D_X}, \boldsymbol{C_X}). \tag{5}$$

The dispatch tensor $\boldsymbol{D_X}$ and the combine tensor $\boldsymbol{C_X}$ are jointly referred to as the **routing tensors**.

---

## 2 MoE layers instantiated by different routers

In this section, we present a family of MoE layers with different underlying routers.

### 2.1 Softmax Token Choice router

We first revisit the Softmax Token Choice in Section 1.1, now in a minibatch setting. To begin, we build a softmax affinity matrix that represents the similarity between each token–expert pair:

$$\boldsymbol{\Pi}_{\text{softmax}} := \text{softmax}\left(\boldsymbol{X}\boldsymbol{W} + \sigma \boldsymbol{\epsilon}\right) \in \mathbb{R}^{T \times E}, \tag{6}$$

where the softmax is applied to each row (normalized across experts). We next use the affinity matrix $\boldsymbol{\Pi}_{\text{softmax}}$ to allocate the routing tensors $\boldsymbol{D_X}$ and $\boldsymbol{C_X}$. Since experts have a fixed buffer capacity $C$, the dispatch tensor $\boldsymbol{D_X}$ and the combine tensor $\boldsymbol{C_X}$ must be allocated in a way such that each expert receives at most $C$ tokens. We achieve this by sequentially going through the rows of affinity matrix $\boldsymbol{\Pi}_{\text{softmax}}$ and assigning each token to its top-scored expert as long as the chosen expert's buffer is not full (i.e., below the buffer capacity $C$). This procedure is called **Token Choice allocation** and is described in Algorithm 1.

The forward pass of the Softmax Token Choice router is summarized below.

---

**Softmax Token Choice router**

1. Compute the token–expert affinity matrix $\boldsymbol{\Pi}_{\text{softmax}}$ in (6) on a batch of input tokens $\boldsymbol{X}$.

2. Use $\boldsymbol{\Pi}_{\text{softmax}}$ to allocate the routing tensors $\boldsymbol{D_X}$ and $\boldsymbol{C_X}$ through Algorithm 1

---

**Balancing expert usage.** In practice, it is beneficial to regularize the parameters $\boldsymbol{W}$ for a balanced usage of experts. Without any regularization on $\boldsymbol{W}$, the top scores at each row of the affinity matrix $\boldsymbol{\Pi}_{\text{softmax}}$ in (6) may concentrate on a few column indices. These indices correspond to certain popular experts that many tokens prefer to choose. But since each expert has a fixed buffer capacity, popular experts may drop most of the tokens in the allocation procedure in Algorithm 1, thus deteriorating the training performance. To prevent this from happening, previous work considered different **auxiliary losses** on $\boldsymbol{W}$. Details of these losses are provided in Appendix A for reference.

---

**Algorithm 1:** Token Choice allocation

---

**Data:**

    (i)   tokens $\boldsymbol{X} \in \mathbb{R}^{T \times D}$

    (ii)  token–expert affinity matrix $\boldsymbol{\Pi} \in \mathbb{R}^{T \times E}$

    (iii) buffer capacity $C \in \mathbb{N}$

    (iv) number of selected experts $k \in \mathbb{N}$ per token (typically $k = 1$ or $2$)

**Result:**

    The routing tensors $\boldsymbol{D_X}$ and $\boldsymbol{C_X} \in \mathbb{R}^{T \times E \times C}$ for the MoE layer (2)

Initialize $\boldsymbol{D_X} \in \mathbb{R}^{T \times E \times C}$ and $\boldsymbol{C_X} \in \mathbb{R}^{T \times E \times C}$ as zero tensors.

**for** the top $i$-th choice with $i = 1, \ldots, k$ **do**

    **for** token index $t = 1, \ldots, T$ **do**

        $w, r = \max_{i\text{-th}} \boldsymbol{\Pi}[t, :]$                 // The value and index of the $i$-th selected score.

        $c = \left\| \boldsymbol{D_X}[:, r, :] \right\|_0$            // The number of tokens already dispatched to the $r$-th expert

        **if** $c < C$ **then**

            $\boldsymbol{D_X}[t, r, c] = 1$

            $\boldsymbol{C_X}[t, r, c] = w.$

---

## 2.2 Sinkhorn Token Choice router

To perform well, the Softmax Token Choice router requires auxiliary losses that balance the expert usage. To eliminate the need for auxiliary losses, recent work (Kool et al., 2021; Clark et al., 2022) proposed the **Sinkhorn Token Choice** router.

Specifically, a softmax affinity matrix $\boldsymbol{\Pi}_{\text{softmax}} = \mathsf{softmax}(\boldsymbol{XW}) \in \mathbb{R}^{T \times E}$ in (6) can be seen as the solution to an entropy-regularized optimization problem:

$$\boldsymbol{\Pi}_{\text{softmax}} = \arg \max_{\boldsymbol{\Pi}} \left[ \langle \boldsymbol{\Pi}, \boldsymbol{XW} \rangle - \langle \boldsymbol{\Pi}, \log \boldsymbol{\Pi} \rangle \right],$$
$$\text{subject to} \begin{cases} \boldsymbol{\Pi} > 0, \\ \boldsymbol{\Pi} \mathbf{1}_E = \mathbf{1}_T, \end{cases} \tag{7}$$

where $\mathbf{1}_T$ is a column vector of size $T$ with values 1. This entropy-regularized characterization decouples the underlying optimization problem and the constraints imposed on the solution. The constraints in Equation (7) are that (i) all token–expert affinity scores have positive values and (ii) the token–expert affinity matrix has a unit row sum, meaning that the affinity scores sum to 1 along the expert axis.

The idea of Sinkhorn Token Choice is to add more constraints to (7). As discussed in Section 2.1, unbalanced expert usage of Softmax Token Choice occurs when top token–expert affinity scores concentrate on a few columns of the affinity matrix $\boldsymbol{\Pi}_{\text{softmax}}$. To promote a balanced expert usage, the Sinkhorn Token Choice router thus requires that all columns are normalized, leading to the following entropy-regularized optimal transport formulation

$$\boldsymbol{\Pi}_{\text{ent}} := \arg \max_{\boldsymbol{\Pi}} \left[ \langle \boldsymbol{\Pi}, \boldsymbol{XW} \rangle - \langle \boldsymbol{\Pi}, \log \boldsymbol{\Pi} \rangle \right],$$
$$\text{subject to} \begin{cases} \boldsymbol{\Pi} > 0, \\ \boldsymbol{\Pi} \mathbf{1}_E = \mathbf{1}_T, \\ \boldsymbol{\Pi}^\top \mathbf{1}_T = (T/E)\mathbf{1}_E. \end{cases} \tag{8}$$

The optimization problem (8) can be solved by Sinkhorn's algorithm (Sinkhorn, 1964; Cuturi, 2013; Peyré & Cuturi, 2019), which iterates between row-wise and column-wise normalization. We then allocate the routing tensors using entropy-regularized transportation plan $\boldsymbol{\Pi}_{\text{ent}}$. In principle, the same $\boldsymbol{\Pi}_{\text{ent}}$ can be used

for dispatching $\boldsymbol{D_X}$ and combining tensors $\boldsymbol{C_X}$. But it is empirically more beneficial[1] to **only allocate** the dispatch tensor $\boldsymbol{D_X}$ using the transportation plan $\boldsymbol{\Pi}_{\text{ent}}$, while still allocating the combine tensor $\boldsymbol{C_X}$ using the softmax matrix $\boldsymbol{\Pi}_{\text{softmax}} = \mathsf{softmax}(\boldsymbol{XW})$. This approach has the benefit that the backward pass of gradient-based training does not go through the Sinkhorn algorithm, which results in faster and more stable training. We provide a case study of this in Figure 3 of the results section; while Clark et al. (2022) recommend using $k = 1$, we have also experimented with $k = 2$ and observe consistent results.

The forward pass of a Sinkhorn Token Choice router is summarized as follows.

---

**Sinkhorn Token Choice router**

1. Compute the token–expert affinity matrix $\boldsymbol{\Pi}_{\text{ent}}$ in (8) on a batch of input tokens $\boldsymbol{X}$.

2. Use $\boldsymbol{\Pi}_{\text{ent}}$ to allocate the dispatch tensors $\boldsymbol{D_X}$ through Algorithm 1

3. Use $\boldsymbol{\Pi}_{\text{softmax}} = \mathrm{softmax}(\boldsymbol{XW})$ to allocate the combine tensor $\boldsymbol{C_X}$ through Algorithm 1

---

### 2.3 Softmax Expert Choice router

Both Softmax Token Choice and Sinkhorn Token Choice share the router allocation algorithm in Algorithm 1. This allocation approach lets each token choose its top-scored expert. However, a limitation of this approach is that it can result in **underused** experts, which are experts that use fewer tokens than their allowed capacity. This can be seen in the inner for-loop of Algorithm 1. There, the number of tokens taken by each expert is restricted to not exceed the buffer capacity $C$, that is, $\left\| \boldsymbol{D_X}[:, r, :] \right\|_0 \leq C$ for each expert $r$. When the inequality is strict, $\left\| \boldsymbol{D_X}[:, r, :] \right\|_0 < C$, the expert $r$ is underused as it could have been possible to allocate more tokens to that expert.

To address the underusage of experts in token-choice allocation, an alternative approach proposed in Zhou et al. (2022) is the **Expert Choice** allocation. In this approach, each expert chooses a fixed amount of tokens. Specifically, given a token–expert affinity matrix $\boldsymbol{\Pi}$, we sequentially go through the columns of $\boldsymbol{\Pi}$ and assign each expert to its top-scored tokens. This procedure is outlined in Algorithm 2.

---

**Algorithm 2:** Expert Choice allocation

**Data:**
   (i)   tokens $\boldsymbol{X} \in \mathbb{R}^{T \times D}$
   (ii)  token–expert affinity matrix $\boldsymbol{\Pi} \in \mathbb{R}^{T \times E}$
   (iii) buffer capacity $C \in \mathbb{N}$

**Result:**
   The routing tensors $\boldsymbol{D_X}$ and $\boldsymbol{C_X} \in \mathbb{R}^{T \times E \times C}$ for the MoE layer (2)

Initialize $\boldsymbol{D_X} \in \mathbb{R}^{T \times E \times C}$ and $\boldsymbol{C_X} \in \mathbb{R}^{T \times E \times C}$ as zero tensors.
**for** expert index $r = 1, \ldots, E$ **do**
   **for** capacity index $c = 1, \ldots, C$ **do**
      $w, t = \max_{c\text{-th}} \left[ \mathsf{Gate}_r(\boldsymbol{x}_1), \ldots, \mathsf{Gate}_r(\boldsymbol{x}_T) \right]$       // The value and index of the $i$-th selected gates.
      $\boldsymbol{D_X}[t, r, l] = 1$
      $\boldsymbol{C_X}[t, r, l] = w$.

---

The Expert Choice allocation in Algorithm 2 offers several benefits compared to the Token Choice allocation in Algorithm 1. Firstly, it ensures all experts receive the same amount of tokens, preventing any from being over or under-used. Secondly, since experts independently select their tokens, many experts can process a single token. This contrasts with Token Choice allocation, where each token is typically assigned to only

---

[1]Personal communications with the authors of Clark et al. (2022).

$k = 1$ or $k = 2$ experts. Such diverse expert participation enhances MoE layer performance (Zhou et al., 2022). Moreover, in the Expert Choice allocation, certain tokens might not be chosen by any experts.

In Zhou et al. (2022), a softmax token–expert affinity matrix $\mathbf{\Pi}_{\text{softmax}} = \mathsf{softmax}(\boldsymbol{XW}) \in \mathbb{R}^{T \times E}$ is used in the Expert Choice allocation of Algorithm 2. We refer it to as the Softmax Expert Choice router, whose procedural steps are summarized as follows.

---

**Softmax Expert Choice router**

1. Compute the token–expert affinity matrix $\mathbf{\Pi}_{\text{softmax}}$ in (6) on a batch of input tokens $\boldsymbol{X}$.

2. Use $\mathbf{\Pi}_{\text{softmax}}$ to allocate the routing tensors $\boldsymbol{D_X}$ and $\boldsymbol{C_X}$ through Algorithm 2

---

### 2.4 Sinkhorn Expert Choice router

We now introduce the **Sinkhorn Expert Choice** router, which fits naturally with the unified MoE layer formulation in Section 1.2. To our knowledge, it has not been previously explored in literature.

Recall that the Softmax Expert Choice router offers versatile expert usage, but it can skip numerous tokens. On the one hand, skipping tokens can be an advantage of Softmax Expert Choice, as the buffer capacity saved by skipped tokens can be allocated to remaining tokens, allowing them to be processed by many experts. On the other hand, skipping too many tokens may lead to inferior results.

To strike a balance between the heterogeneous usage of experts and the goal of minimizing token dropping, we can use the entropy-regularized optimal transport plan $\mathbf{\Pi}_{\text{ent}}$ to allocate the routing tensors through Algorithm 2. As introduced in Section 2.2, the optimal transport plan $\mathbf{\Pi}_{\text{ent}}$ has normalized column sums. Intuitively, while $\mathbf{\Pi}_{\text{ent}}$ reduces token dropping, it still allows a variable number of experts applied to tokens. The procedure of this Sinkhorn Expert Choice router is summarized as follows.

---

**Sinkhorn Expert Choice router**

1. Compute the token–expert affinity matrix $\mathbf{\Pi}_{\text{ent}}$ in (8) on a batch of input tokens $\boldsymbol{X}$.

2. Use $\mathbf{\Pi}_{\text{ent}}$ to allocate the dispatch tensor $\boldsymbol{D_X}$ through Algorithm 2

3. Use $\mathbf{\Pi}_{\text{softmax}} = \mathrm{softmax}(\boldsymbol{XW})$ to allocate the combine tensor $\boldsymbol{C_X}$ through Algorithm 2

---

Note that, as in the Sinkhorn Token Choice, we only allocate the dispatch tensor $\boldsymbol{D_X}$ using $\mathbf{\Pi}_{\text{ent}}$ while calculating the combine tensor using $\boldsymbol{C_X}$ using $\mathbf{\Pi}_{\text{softmax}}$.

### 2.5 Sparsity-constrained Expert Choice router

All MoE layers introduced so far use specific sorting heuristics to allocate routing tensors. The sorting is applied either to the expert axis, as in Algorithm 1, or the token axis, as in Algorithm 2, to create sparse tensors $\boldsymbol{D}$. This is necessary because the token–expert affinity matrix $\mathbf{\Pi}$ is neither sparse nor constrained by the buffer capacity.

An alternative approach is to promote sparsity in $\mathbf{\Pi}$ to directly meet the buffer capacity constraint (Liu et al., 2023). This is a more principled way to create a sparse allocation of experts. To this end, we consider a token–expert affinity matrix $\mathbf{\Pi}_{\text{sparse}}$ that solves a sparsity–constrained optimal transport problem:

$$\mathbf{\Pi}_{\text{sparse}} := \arg\max_{\mathbf{\Pi}} \ \left[ \langle \mathbf{\Pi}, \mathsf{softmax}(\boldsymbol{XW}) \rangle - \frac{1}{2} \|\mathbf{\Pi}\|_{\text{F}}^2 \right],$$

$$\text{subject to} \begin{cases} \mathbf{\Pi} \geq 0, \\ \mathbf{\Pi}\mathbf{1}_E = \mathbf{1}_T, \\ \mathbf{\Pi}^\top \mathbf{1}_T = (T/E)\mathbf{1}_E \\ \|\mathbf{\Pi}[:, r]\|_0 \leq C \text{ for all } r \in [E]. \end{cases} \tag{9}$$

There are several key differences between the sparsity–constrained optimal transport in (9) and the entropy-regularized optimal transport in (8). First, sparsity–constrained optimal transport in (9) uses a quadratic regularization $\|\mathbf{\Pi}\|_{\mathrm{F}}^2$ instead of the entropy regularization $\langle \mathbf{\Pi}, \log \mathbf{\Pi} \rangle$ on the transportation plan $\mathbf{\Pi}$. This quadratic regularization, as shown in Blondel et al. (2018), preserves the sparsity in the transportation plan and allows for an efficient numerical solver. Second, the formulation (9) introduces an additional constraint to upper bound the number of nonzeros of each column of $\mathbf{\Pi}$ by the buffer constraint $C$. This ensures that all experts use up to $C$ tokens. Third, in the first term of the sparsity–constrained optimal transport in (9), we evaluate the inner product of $\mathbf{\Pi}$ with a positive utility matrix $\mathsf{softmax}(\boldsymbol{XW})$ as opposed to the raw matrix product $\boldsymbol{XW}$ in (8). The positive utility matrix encourages each column of $\mathbf{\Pi}$ to have exactly $C$ nonzeros: even though we only upper bound the number of nonzeros in each column of $\mathbf{\Pi}$ by $C$ in (9), a learned optimal transport plan $\mathbf{\Pi}_{\mathrm{sparse}}$ will usually take exactly $C$ nonzeros per column since it needs to maximize its inner product.

Finding an exact minimizer $\mathbf{\Pi}_{\mathrm{sparse}}$ in (9) can be intractable in general due to the non-convexity of this constrained optimization problem. Nonetheless, an approximate $\mathbf{\Pi}_{\mathrm{sparse}}$ can be obtained using a semi-dual or a dual formulation; for more details refer Liu et al. (2023). The overall procedure of the sparsity-constrained MoE layer is summarized as follows.

---

**Sparsity-constrained router**

1. Approximate the token–expert affinity matrix $\mathbf{\Pi}_{\mathrm{sparse}}$ in (9) on a batch of input tokens $\boldsymbol{X}$.

2. Use $\mathbf{\Pi}_{\mathrm{sparse}}$ to allocate the dispatch tensor $\boldsymbol{D_X}$ through Algorithm 2

3. Use $\mathbf{\Pi}_{\mathrm{softmax}} = \mathsf{softmax}(\boldsymbol{XW})$ to allocate the combine tensor $\boldsymbol{C_X}$ through Algorithm 2

---

### 2.6 Soft MoE

All MoE formulations we have introduced so far aim to find **hard assignments** between tokens and experts; these hard assignments are represented by the binary, $\{0,1\}$-valued dispatch tensors $\boldsymbol{D_X}$ in token choice allocation (Algorithm 1) and expert choice allocation (Algorithm 2). In these formulations, each expert can process either an entire token or none of it. In a different approach, Soft MoEs (Puigcerver et al., 2023) instead allow experts to process weighted combinations of tokens, offering more flexibility.

We now present Soft MoE using our unified formulation of MoE layers (2). One key feature of SoftMoE is that we define $T$ as the token count in a single image, since tokens are linearly combined exclusively from the same input. This contrasts with other routers we discussed, where $T$ represented the number of tokens in a batch of multiple inputs.

We let $\boldsymbol{\Phi} \in \mathbb{R}^{D \times E \times C}$ be a learnable tensor, where each vector $\boldsymbol{\Phi}[:, r, c] \in \mathbb{R}^D$ represents the features of the $c$-th buffer slot of the $r$-th expert. For input $\boldsymbol{X} \in \mathbb{R}^{T \times D}$, we compute the affinity scores between all tokens and the $(r, c)$-th slot with:

$$\boldsymbol{Z}[:, r, c] := \boldsymbol{X}\boldsymbol{\Phi}[:, r, c] \in \mathbb{R}^T. \tag{10}$$

The Soft MoE approach then computes the routing tensors $\boldsymbol{D_X}$ and $\boldsymbol{C_X}$ by normalizing the affinity scores in $\boldsymbol{Z} \in \mathbb{R}^{T \times E \times C}$:

$$\boldsymbol{D_X}[t, r, c] = \frac{\exp \boldsymbol{Z}[t, r, c]}{\sum_{t'} \exp \boldsymbol{Z}[t', r, c]}, \quad \boldsymbol{C_X}[t, r, c] = \frac{\exp \boldsymbol{Z}[t, r, c]}{\sum_{r', c'} \exp \boldsymbol{Z}[t, r', c']}. \tag{11}$$

Here, the dispatch tensor $\boldsymbol{D_X}$ is normalized along the token axis: we dispatch fractions of every token to each slot, with the fractions summing up to 1. Similarly, the combine tensor $\boldsymbol{C_X}$ is normalized along the slot dimensions $(r, c)$, making each MoE output token a convex combination of all slot outputs with combining weights summing up to 1. We summarize the forward pass of Soft MoE as follows.

---

**Soft MoE router**

1. Compute the affinity tensor $\boldsymbol{Z} \in \mathbb{R}^{T \times E \times C}$ for tokens and expert slots via (10).

2. Compute the routing tensors $\boldsymbol{D_X}$ and $\boldsymbol{C_X}$ in (11).

---

Note that the routing tensors $\boldsymbol{D_X}$ and $\boldsymbol{C_X}$ differ from previous approaches by not being binary or sparse. Nonetheless, Soft MoEs are computationally efficient. This is because each expert only processes a fixed number of slots per input that is significantly smaller than the sequence length. In addition, compared to other methods, Soft MoEs have a speed advantage as they avoid sorting and optimal transport algorithms.

## 3 Experiments

We report comprehensive comparisons of vision MoE models with the 6 routers we introduced earlier: (1) Softmax Token Choice, (2) Sinkhorn Token Choice, (3) Softmax Expert Choice, (4) Sinkhorn Expert Choice, (5) Sparsity-constrained Expert Choice, and (6) Soft MoE router. We evaluate MoE models using both large-scale pre-training and few-shot adaptation experiments.

### 3.1 Models

As in earlier vision MoE work (Riquelme et al., 2021), we replace a subset of the dense feedforward layers in a vision transformer (Dosovitskiy et al., 2021) with MoE layers. Specifically, we use the Every-2 variant in Riquelme et al. (2021) that places the MoEs on every other layer. For the comparison to be comprehensive, we use models of different sizes with a naming convention consistent to vision transformer: B(ase)32, B(ase)16, and L(arge)16. The numbers 32 and 16 here refer to $32 \times 32$ and $16 \times 16$ patch sizes. Models that use a smaller patch size result in more patches, and thus model B16 has a greater capacity and computational cost than model B32. We fix the total number of experts to be 32; that is, $E = 32$.

For each architecture (B32, B16, and L16) and each router, we experiment with two configurations to experiment with different capacity $C$:

- For Softmax Token Choice and Sinkhorn Token Choice routers that process each token with $k$ experts, we experiment with $k = 1$ and $k = 2$. In this way, the buffer capacity (the number of tokens an expert can process at most in a batch) of these variants is $C = \text{round}(k \cdot T/E)$.

- For Softmax Expert Choice, Sinkhorn Expert Choice, and sparsity-constrained variants, we control the buffer capacity $C$ through a capacity factor $c$, which plays a role similar to $k$. The buffer capacity $C$ is defined through $C = \text{round}(c \cdot T/E)$. We experiment $c$ with 1 or 2, which match the buffer capacity of $k = 1$ and $k = 2$ in the Token Choice cases.

- For Soft MoE routers, we set the number of slots per expert to be $C = \text{round}(c \cdot T/E)$ with $c = 1$ or 2, just like in the Expert Choice case.

### 3.2 Upstream task results: Pre-training on JFT-300M

For the pre-training experiments, all models were trained on the JFT-300M (Sun et al., 2017), which contains about 305 million training images and 50,000 validation images, organized in a hierarchy of 18,291 different classes. To avoid overlap with the validation and test sets of JFT-300M, the images in the dataset were deduplicated, as done in Kolesnikov et al. (2020). Our main metric for JFT-300 is the top-1 classification accuracy (Prec@1).

**Accuracy comparison.** Our pretraining results, shown in Figure 1. In terms of accuracy (y-axis), the Expert Choice variants (■, ●, ▲) generally surpass Token Choice variants (✖, ✚) in accuracy. Among Token Choice models, those using a Sinkhorn-based token-expert affinity matrix (✚) perform better than those with a Softmax-based matrix (✖), aligning with previous findings in language MoE research (Clark et al., 2022).

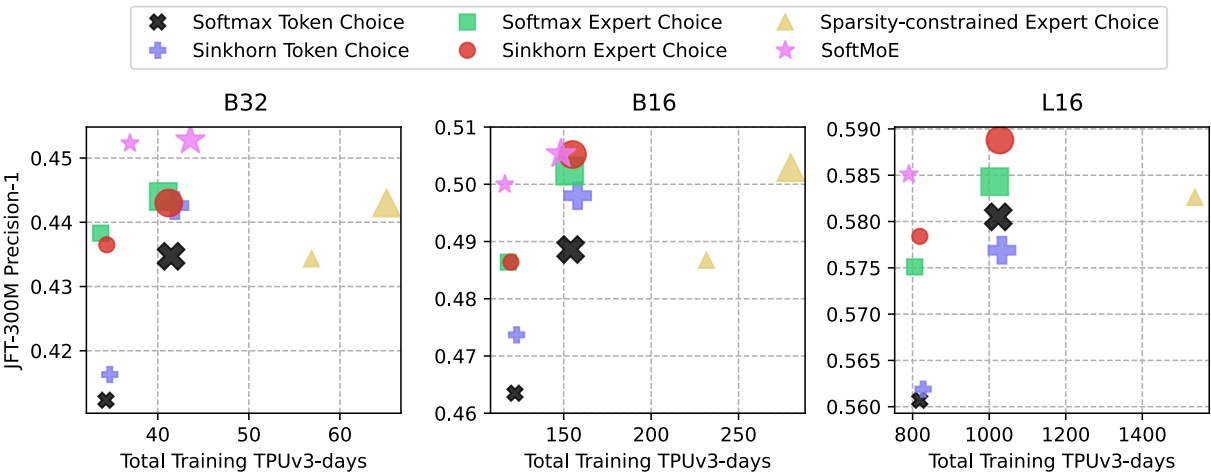

Figure 1: **Comparison of training time and performance in the JFT300M dataset for image classification.** The marker size represents the router's capacity, with smaller and larger sizes indicating lower and higher capacities.

However, in Expert Choice models, the choices of token-expert affinity matrix—Softmax (■), Sinkhorn (●), or sparsity-constrained (▲)—show less significant impact on performance. We hypothesize that Sinkhorn transform effectively balances the expert usage in the Token Choices case, thus improving the performance relative to the Softmax Token Choice; but such a balance is no longer needed for the Expert Choice cases, and therefore its impact is small. Compared to Token Choice and Expert Choice routers, SoftMoE routers (★) outperform both by a noticeable margin.

There is one instance of SoftMoE for L-16 at $c = 2$ not included in Figure 1 (one ★ is missing from the right-most panel of Figure 1). This is due to training instability that occurred at that specific instance. This might relate to using only 32 experts ($E = 32$) for consistent comparisons across all routers. Notably, SoftMoE performs best with a larger number of experts (e.g., $E = 2048$) and few slots per expert, as shown in Puigcerver et al. (2023, Figure 6). Despite not being the most optimal setup, SoftMoE surpasses other configurations in Figure 1, where instability was not an issue.

**Training cost comparison.** Considering the computational cost, the sparsity-constrained router (▲) is more expensive compared to others, primarily because of its sorting and sparse projection steps (Liu et al., 2023). In contrast, SoftMoE routers have the best compute–accuracy tradeoff.

For a comprehensive numerical comparison of routers using the JFT300M dataset, refer to Tables 1, 2, and 3 in Appendix B.

### 3.3 Downstream task results: Few-shot transfer on ImageNet-1k

To assess how well pre-trained MoE models adapt to new tasks, we conducted few-shot adaptation experiments using the ImageNet-1k dataset (Deng et al., 2009). In these experiments, we used 10 image samples per class from ImageNet-1k. The pre-trained model extracts a fixed feature embedding for each image, which is then used to train a linear regression model. This linear model maps the extracted features to the one-hot encoded target labels. This procedure is in line with the 10-shot evaluation procedure described by Dosovitskiy et al. (2021); Riquelme et al. (2021).

Our few-shot results, as shown in Figure 2, confirm findings from the pretraining phase. Overall, the SoftMoE router (★) consistently outperformed the Expert Choice routers (■, ●, ▲), which themselves surpassed the performance of the Token Choice routers (✖, ✚). In most scenarios, token choice routers marked by ✚ that use a Sinkhorn-derived token-expert affinity matrix tend to outperform those using a Softmax-based affinity matrix (✖). However, the performance differences between the Expert Choice routers (■, ●, ▲), are relatively

minor. This emphasizes that the choice of routing tensor allocation algorithm is more crucial than the specific method of parameterizing the token-expert affinity matrix.

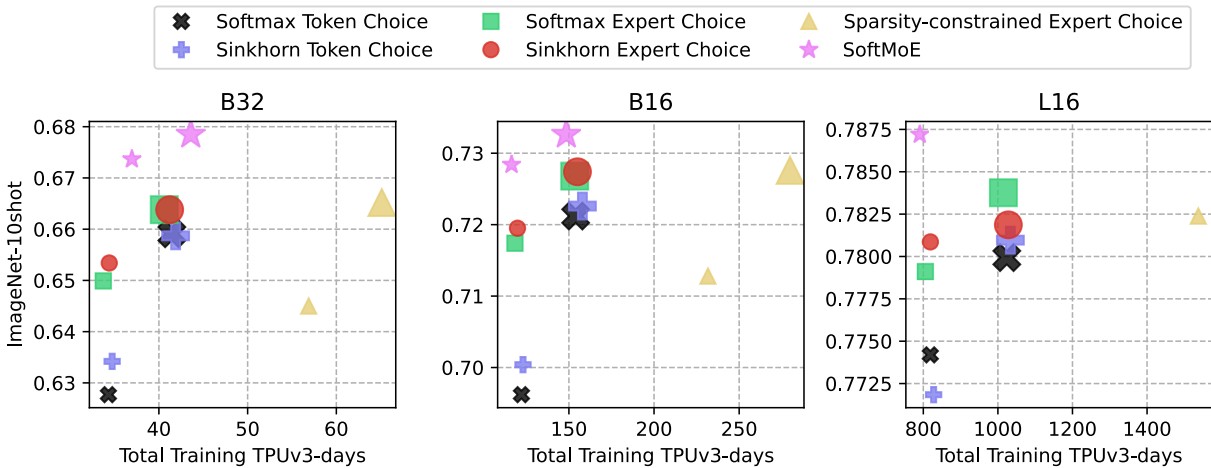

Figure 2: **Comparison of training time and performance in a 10-shot Transfer Task on the ImageNet-1k Dataset.** The marker size represents the router's capacity, with smaller and larger sizes indicating lower and higher capacities.

For a comprehensive numerical comparison of routers few-shot adapted on the ImageNet-1k dataset, refer to Tables 1, 2, and 3 in Appendix B.

### 3.4 Analysis: the usage of Softmax combine tensors in Sinkhorn routers.

In Section 2.2, we mentioned a useful trick in the Sinkhorn Token Choice router: using the Sinkhorn-based matrix $\mathbf{\Pi}_{\text{ent}}$ to allocate the dispatch tensor $\boldsymbol{D_X}$ and the Softmax-based matrix $\mathbf{\Pi}_{\text{softmax}}$ for the combine tensor $\boldsymbol{C_X}$. This approach, also applied in Sinkhorn Expert Choice and Sparsity-constrained Expert Choice routers, primarily aims to boost processing speed by bypassing the optimal transport algorithm in the backward pass. Perhaps surprisingly, it also enhances performance. Figure 3 showcases a Sinkhorn Token Choice case study. We contrast two methods: one assigns $\boldsymbol{D_X}$ with $\mathbf{\Pi}_{\text{ent}}$ and $\boldsymbol{C_X}$ with $\mathbf{\Pi}_{\text{softmax}}$ (labeled "with softmax combine"), while the other uses $\mathbf{\Pi}_{\text{ent}}$ for both tensors (labeled "without softmax combine"). The "with softmax combine" variant achieves approximately 1% higher accuracy on the JFT300M dataset.

## 4 Related work

MoE models have been increasingly used in language and vision domains. We briefly review the application of them in this section. We refer the interested readers to Fedus et al. (2022a) for a comprehensive review.

**MoEs for language.** MoEs have been successfully applied to language modeling and machine translation. One of the earliest successes of sparse MoEs in language modeling and machine translation was demonstrated by Shazeer et al. (2017). They insert MoE layers between LSTM layers (Hochreiter & Schmidhuber, 1997) to increase the model capacity while maintaining high computational efficiency. This approach achieved state-of-the-art at that time, with a lower computational cost than baseline models. Sparse MoEs have further advanced language modeling when combined with Transformers (Vaswani et al., 2017). The GShard (Lepikhin et al., 2021) and Switch Transformers (Fedus et al., 2022b) are among the earliest works that replace feed-forward layers in Transformers with sparse MoE layers. In addition, research efforts have been made to analyze and simplify the routers in language MoEs, such as deterministic routers (Lewis et al., 2021) and routers based on reinforcement learning and optimal transport (Kool et al., 2021; Clark et al., 2022).

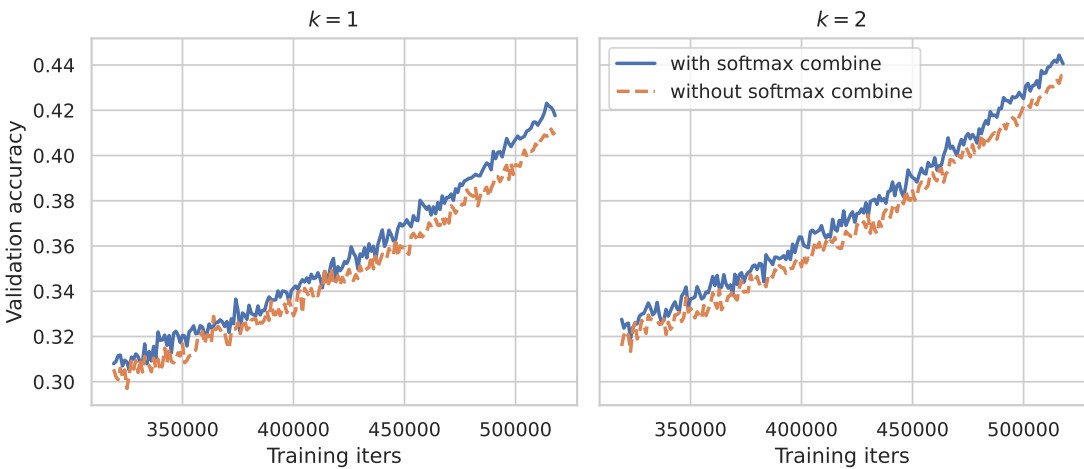

Figure 3: **Assessing the impact of using Softmax-based combine tensors in Sinkhorn Token Choice routers.** The number of selected expets are $k = 1$ (left panel) and $k = 2$ (right panel). Both routers are used in a B32 architecture. The $x$ axis shows the training iteration number, while the $y$ axis shows the validation accuracy on the JFT300M dataset.

**MoEs for vision.** In the realm of vision, early work on MoEs (Eigen et al., 2013; Ahmed et al., 2016; Gross et al., 2017; Abbas & Andreopoulos, 2020; Wang et al., 2020; Pavlitskaya et al., 2020; Yang et al., 2019) were mostly based on convolutional neural networks and were applied to specific tasks. The tremendous success of Vision Transformers (ViTs) (Dosovitskiy et al., 2021) has motivated research in creating sparse MoEs based on ViTs. Riquelme et al. (2021) introduced V-MoE, a vision MoE architecture based on ViTs, and showed that it outperforms dense ViT counterparts at the same computational cost. Beyond accuracy, vision MoEs have been shown to offer better robustness against adversarial attacks (Puigcerver et al., 2022). Furthermore, while (Riquelme et al., 2021; Puigcerver et al., 2022) focus on image classification problems, Wu et al. (2022) show the capability of vision MoEs to solve high-resolution vision tasks such as segmentation and detection. Li et al. (2022) proposed a variant of vision MoE with enhanced domain-generalization abilities. Puigcerver et al. (2023) formulated soft MoEs that are fully differentiable while being as efficient as sparse MoEs.

**MoEs for multimodal and multitask learning.** Mustafa et al. (2022) presented the first multimodal sparse MoE model, called Language-Image MoE (LIMoE). LIMoE processes both images and text in a modality-agnostic fashion, to align image and text embeddings via contrastive learning. LIMoE matches the performance of state-of-the-art dense models and surpasses dense baselines at equivalent computational costs. MoEs have also been used in multitask settings (Ma et al., 2018; Chen et al., 2023).

**Other aspects of MoEs.** There are a few research studies on MoEs from perspectives that are not tied to specific data modalities. For example, Hwang et al. (2023) proposed an efficient pipeline that optimizes the implementation of MoE layers on GPUs. From a theoretical perspective, Chen et al. (2022) studies why simple sparse MoEs do not collapse into a single model. Komatsuzaki et al. (2023) create sparse MoEs from pre-existing dense models as a way to reuse the sunk cost for training dense models. This simple yet powerful approach can be generally applied to tasks in different modalities.

## 5 Conclusion

MoEs offer a promising solution to large-scale machine learning applications. Our paper presents the first comprehensive study of transformer-based sparse and soft MoEs in computer vision tasks, achieved through a unified MoE layer formulation with routers. We show that the strong performance of many language

MoEs carries over to vision. We found routing tensor allocation crucial for sparse MoEs, more so than other factors. Notably, soft MoEs outperform all sparse alternatives tested.

Research on efficient sparse MoEs for vision problems at scale is still in its early stages. As MoEs can handle large amounts of data while keeping computational costs low, it is expected that they will become increasingly important for data-rich tasks in the future. Understanding how different MoE models perform in these tasks is crucial. Our paper takes a step in this direction and opens new opportunities for further study of MoEs at scale.

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

## A   Auxiliary losses for routers

**Importance loss.**   The importance loss penalizes the variability of each expert's overall gating weights on a minibatch of tokens. Let $\boldsymbol{X} \in \mathbb{R}^{T \times E}$ be a minibatch of $T$ tokens, where each token is a row of $\boldsymbol{X}$. We let

$$\boldsymbol{\Pi}_{\boldsymbol{W},\boldsymbol{X}} = \mathsf{softmax}\big(\boldsymbol{X}\boldsymbol{W}\big) \in \mathbb{R}^{T \times E}$$

be a matrix token-to-experts routing probabilities, where the $\mathsf{softmax}$ operator applies to each row of $\boldsymbol{X}\boldsymbol{W}$. Summing over rows, the vector

$$\boldsymbol{r}_{\boldsymbol{W},\boldsymbol{X}} := \boldsymbol{\Pi}_{\boldsymbol{W},\boldsymbol{X}}^{\top} \mathbf{1}_T \in \mathbb{R}^{E}$$

describe each expert's token-receiving probabilities summed on that minibatch of tokens. Since we want each expert to receive roughly the same amount of tokens, the token-receiving probabilities of experts should not vary considerably. To minimize the variation, the importance loss is defined as

$$\mathcal{L}_{\mathsf{Imp}}(\boldsymbol{X}, \boldsymbol{W}) = \widehat{\mathrm{CV}}\Big(\big\{\boldsymbol{r}_{\boldsymbol{W},\boldsymbol{X}}[i]\big\}_{i=1}^{E}\Big)^2,$$

where $\widehat{\mathrm{CV}}$ denotes the empirical coefficient of variation.

**Load loss.**   The load loss of a vision MoE has the form

where $\delta$ is a real-valued random variable with a Gaussian distribution $\mathcal{N}(0, \frac{1}{E})$.

$$\mathrm{load}_i(\boldsymbol{x}, \boldsymbol{W}, \boldsymbol{\epsilon}) := \mathbb{P}_\delta\Big(\big(\boldsymbol{W}\boldsymbol{x}\big)[i] + \delta > \max_{k\text{-th}}\big(\boldsymbol{W}\boldsymbol{x} + \boldsymbol{\epsilon}\big)\Big) \tag{12}$$

$$= \mathbb{P}_\delta\Big(\delta > \max_{k\text{-th}}\big(\boldsymbol{W}\boldsymbol{x} + \boldsymbol{\epsilon}\big) - \big(\boldsymbol{W}\boldsymbol{x}\big)[i]\Big) \tag{13}$$

$$= \Phi\Big(\big(\boldsymbol{W}\boldsymbol{x}\big)[i] - \max_{k\text{-th}}\big(\boldsymbol{W}\boldsymbol{x} + \boldsymbol{\epsilon}\big)\Big), \tag{14}$$

where $\Phi$ is the cumulative distribution function of the Gaussian distribution $\mathcal{N}(0, \frac{1}{E})$

Summing the loads (12) over all tokens in $\boldsymbol{X}$ yields the expected numbers of tokens received for each expert. We define the **load loss** as the coefficient of variation of such expected numbers of received tokens across experts:

$$\mathcal{L}_{\mathrm{load}}(\boldsymbol{X}; \boldsymbol{W}) = \widehat{\mathrm{CV}}\Big(\big\{\mathrm{load}_i(\boldsymbol{X}; \boldsymbol{W})\big\}_{i=1}^{E}\Big)^2 \quad \text{with } \mathrm{load}_i(\boldsymbol{X}, \boldsymbol{W}) = \sum_{t=1}^{T} \mathrm{load}_i(\boldsymbol{x}^{(t)}; \boldsymbol{W}, \boldsymbol{\epsilon}^{(t)}). \tag{15}$$

## B   Detailed experimental results

|  | Accuracy | Training TPUv3-days |
|---|---|---|
| Softmax ($k = 1$) | 41.23 % | 34.31 |
| Sinkhorn ($k = 1$) | 41.63 % | 34.73 |
| Softmax Expert Choice ($C_{\text{factor}} = 1$) | 43.83 % | 33.73 |
| Sinkhorn Expert Choice ($C_{\text{factor}} = 1$) | 43.65 % | 34.40 |
| Sparsity constrained ($C_{\text{factor}} = 1$) | 43.43 % | 56.88 |
| SoftMoE | **45.23 %** | 36.95 |
| Softmax Token Choice ($k = 2$) | 43.47 % | 41.46 |
| Sinkhorn Token Choice ($k = 2$) | 44.26 % | 41.88 |
| Softmax Expert Choice ($C_{\text{factor}} = 2$) | 44.40 % | 40.60 |
| Sinkhorn Expert Choice ($C_{\text{factor}} = 2$) | 44.30 % | 41.21 |
| Sparsity constrained ($C_{\text{factor}} = 2$) | 44.30 % | 65.13 |
| SoftMoE | **45.28 %** | 43.59 |

Table 1: Comparing routers in B32 architecture using the JFT dataset.

|  | Accuracy | Training TPUv3-days |
|---|---|---|
| Softmax Token Choice ($k = 1$) | 46.35 % | 122.12 |
| Sinkhorn Token Choice ($k = 1$) | 47.37 % | 123.04 |
| Softmax Expert Choice ($C_{\text{factor}} = 1$) | 48.64 % | 118.31 |
| Sinkhorn Expert Choice ($C_{\text{factor}} = 1$) | 48.64 % | 119.80 |
| Sparsity constrained ($C_{\text{factor}} = 1$) | 48.67 % | 231.66 |
| SoftMoE | **50.00 %** | 116.31 |
| Softmax Token Choice ($k = 2$) | 48.86 % | 153.96 |
| Sinkhorn Token Choice ($k = 2$) | 49.80 % | 157.93 |
| Softmax Expert Choice ($C_{\text{factor}} = 2$) | 50.23 % | 153.42 |
| Sinkhorn Expert Choice ($C_{\text{factor}} = 2$) | 50.52 % | 155.03 |
| Sparsity constrained ($C_{\text{factor}} = 2$) | 50.29 % | 279.89 |
| SoftMoE | **50.53 %** | 148.47 |

Table 2: Comparing routers in B16 architecture using the JFT dataset.

|  | Accuracy | Training TPUv3-days |
|---|---|---|
| Softmax Token Choice ($k = 1$) | 56.07 % | 818.80 |
| Sinkhorn Token Choice ($k = 1$) | 56.19 % | 827.70 |
| Softmax Expert Choice ($C_{\text{factor}} = 1$) | 57.51 % | 805.60 |
| Sinkhorn Expert Choice ($C_{\text{factor}} = 1$) | **57.84 %** | 818.95 |
| Sparsity constrained ($C_{\text{factor}} = 1$) | - | - |
| SoftMoE | 58.51% | 790.57 |
| Softmax Token Choice ($k = 2$) | 58.05 % | 1023.92 |
| Sinkhorn Token Choice ($k = 2$) | 57.69 % | 1033.43 |
| Softmax Expert Choice ($C_{\text{factor}} = 2$) | 58.43 % | 1014.71 |
| Sinkhorn Expert Choice ($C_{\text{factor}} = 2$) | **58.88 %** | 1028.06 |
| Sparsity constrained ($C_{\text{factor}} = 2$) | 58.26 % | 1537.41 |
| SoftMoE | - | - |

Table 3: Comparing routers in L16 architecture using the JFT dataset.

|  | Accuracy |
|---|---|
| Softmax Token Choice ($k = 1$) | 62.77 % |
| Sinkhorn Token Choice ($k = 1$) | 63.42 % |
| Softmax Expert Choice ($C_{\text{factor}} = 1$) | 64.99 % |
| Sinkhorn Expert Choice ($C_{\text{factor}} = 1$) | 65.34 % |
| Sparsity constrained ($C_{\text{factor}} = 1$) | 64.50 % |
| SoftMoE | **67.37 %** |
| Softmax Token Choice ($k = 2$) | 65.91 % |
| Sinkhorn Token Choice ($k = 2$) | 65.87 % |
| Softmax Expert Choice ($C_{\text{factor}} = 2$) | 66.38 % |
| Sinkhorn Expert Choice ($C_{\text{factor}} = 2$) | 66.52 % |
| Sparsity constrained ($C_{\text{factor}} = 2$) | 65.75 % |
| SoftMoE | **67.85 %** |

Table 4: Comparing routers in B32 architecture on the ImageNet 10-shot task.

Table 5: B16 on ImageNet 10-shot

|  | Accuracy |
|---|---|
| Softmax Token Choice ($k = 1$) | 69.62 % |
| Sinkhorn Token Choice ($k = 1$) | 70.04 % |
| Softmax Expert Choice ($C_{\text{factor}} = 1$) | 71.74 % |
| Sinkhorn Expert Choice ($C_{\text{factor}} = 1$) | 71.95 % |
| Sparsity constrained ($C_{\text{factor}} = 1$) | 71.28 % |
| SoftMoE | **72.84 %** |
| Softmax Token Choice ($k = 2$) | 72.12 % |
| Sinkhorn Token Choice ($k = 2$) | 72.26 % |
| Softmax Expert Choice ($C_{\text{factor}} = 2$) | 72.68 % |
| Sinkhorn Expert Choice ($C_{\text{factor}} = 2$) | 72.74 % |
| Sparsity constrained ($C_{\text{factor}} = 2$) | 72.76 % |
| SoftMoE | **73.26 %** |

Table 6: Comparing routers in B16 architecture on the ImageNet 10-shot task.

Table 7: L16 on ImageNet 10-shot

|  | Accuracy |
|---|---|
| Softmax Token Choice ($k = 1$) | 77.42 % |
| Sinkhorn Token Choice ($k = 1$) | 77.18 % |
| Softmax Expert Choice ($C_{\text{factor}} = 1$) | 77.91 % |
| Sinkhorn Expert Choice ($C_{\text{factor}} = 1$) | 78.08 % |
| Sparsity constrained ($C_{\text{factor}} = 1$) | - |
| SoftMoE | **78.72 %** |
| Softmax Token Choice ($k = 2$) | 78.00 % |
| Sinkhorn Token Choice ($k = 2$) | 78.10 % |
| Softmax Expert Choice ($C_{\text{factor}} = 2$) | **78.38 %** |
| Sinkhorn Expert Choice ($C_{\text{factor}} = 2$) | 78.19 % |
| Sparsity constrained ($C_{\text{factor}} = 2$) | 78.24 % |
| SoftMoE | - |

Table 8: Comparing routers in L16 architecture on the ImageNet 10-shot task.

