# OpenReview forum: "Routers in Vision Mixture of Experts: An Empirical Study"
_TMLR — Accepted by TMLR_

### Review · Reviewer_VVoS · 2024-02-24

**Summary Of Contributions:**

This paper surveys and measures several current MoE variations in the context of image classification using MoE MLPs insert into ViT.  The factorial combination of (Softmax, Sinkhorn) x (Token choice, Expert choice) is examined, as well as the newer soft MoE.  Each is summarized using the same formal framework, and measured on both JFT-300M classification and downstream ImageNet 10-shot classification.  The results corroborate recent findings that soft MoE outperforms Expert choice, and Expert choice outperforms Token choice, for these settings.

**Audience:**

Yes

**Claims And Evidence:**

No

**Requested Changes:**

* more explicitly describe the unified formulation (esp D, C tensors) as drawn from Puigcerver et al, including in descriptions and the first contribution summary points

* edit descriptions for the second contribution point to be more concrete and suitable to the contributions; at the moment it is a little vague.  to me these seem to be the measurement of Sinkhorn to fill out the factorial combination of (token,expert) x (softmax,sinkhorn), and a corroboration of prior results using JFT-300M, but there may be others

* fix smaller issues mentioned above in review above

* Additional measurements on another dataset or task substantially different from JFT or ImageNet would strengthen this paper considerably, particularly since Puigcerver focuses on JFT.

**Strengths And Weaknesses:**

Overall, these contributions seem small, but may still be of interest.  Measuring on another dataset or task substantially different from JFT would greatly strengthen this paper, as most of the previous works on which this paper is based focus on JFT.

To me, phrasing all of these different experts in the unified framework is useful, but the framework isn't too far away from most MoE descriptions in the first place.  Still, it's nice to see how they all relate surveyed in one place, in the context of ViTs.  I'd note, though, that the dispatch D, combine C is the same as in Puigcerver et al 2023, and that this work rephrases the other MoEs in that framework.  The framework itself seems to be drawn from this prior work.

Most of the main evaluation results are also already examined in Puigcerver et al 2023 "From Sparse to Soft Mixtures of Experts" --- Figs 3 and 22 in Puigcerver show exactly the same measurements comparing Token choice, Expert choice and Soft MoE that this paper describes, but on JFT-4B instead of 300M.  However, this paper fills in additional points of comparison, also looking at Sinkhorn routing in both Token and Expert choice contexts (with this paper being the first comparison using Expert choice context, according to the authors).

Additional questions and typos:

* p.7: is the sign wrong for the $||\Pi||_F^2$ ?  Combined with a constraint that the rows (or cols) sum to 1, a sparser vector maximizes this norm, not minimizes it, as is now mentioned in the text.  Or, it could be that the final constraint $||\Pi[:,r]||_0 \le C$ is what imposes sparsity (use of <= C entries), and the sign on the 2 norm should minimize, not to encourage sparsity, but the opposite, to encourage full, balanced use of all C entries?  Could you please check this description?

* p.9 second bullet: should this read "expert choice"?  both bullet points now say "token choice".

* p.10: text says black x for sparsity-constrained router, should be yellow triangle

---

> ### Author Response · Authors · 2024-03-05
> **Reply from the authors**
>
> Thank you for your constructive reviews! We appreciate your points of critique and we have incorporated them into the revised version; the edited text is highlighted in brown.
>
> >p.7: is the sign wrong for the $||\boldsymbol{\Pi}||_F^2$?
>
> The minus sign is correct here: since the squared Frobenius norm is a regularization term (to be minimized), its sign is the opposite of the first term that we want to maximize. We acknowledge that this might seem unusual because most optimal transport work uses an argmin formulation. We use argmax because we see $\textrm{softmax}(\boldsymbol{X} \boldsymbol{W})$ as an affinity matrix and not a cost matrix. A similar argmax formulation in entropy-regularized optimal transport can be found in Equation (17) in [Clark et al. (2022)](https://arxiv.org/pdf/2202.01169.pdf), where there is a minus sign in front of the regularizer as well.
>
>
> > p.9 second bullet: should this read "expert choice"? both bullet points now say "token choice".
> > p.10: text says black x for sparsity-constrained router, should be yellow triangle
>
> Sorry about these typos. We have corrected them in our newly updated paper.
>
> > $\textbullet{}$ more explicitly describe the unified formulation (esp D, C tensors) as drawn from Puigcerver et al, including in descriptions and the first contribution summary points $\textbullet{}$ edit descriptions for the second contribution point to be more concrete and suitable to the contributions; at the moment it is a little vague. to me these seem to be the measurement of Sinkhorn to fill out the factorial combination of (token,expert) x (softmax,sinkhorn), and a corroboration of prior results using JFT-300M, but there may be others
>
> Thanks for these suggestions, which have helped us define our contributions more clearly. We have updated our manuscript accordingly, and the revisions to our contributions can be found on page 2.

---

> > ### Comment · Reviewer_VVoS · 2024-04-10
> > **comments**
> >
> > Thanks for your comments.  My larger concerns are addressed.
> >
> > For the sign of $||\Pi||^2_F$, I agree it should be a minus sign.  But I don't see yet how this matches the description that it encourages sparsity on the bottom of p7, "This quadratic regularization ... promotes sparsity".  Based on the equations written, it looks like it has the opposite effect, promoting fuller use of all experts, and the constraint $||\Pi[:,r]||_0 \le C$ is what determines sparsity here:  Given the other constraint, $\Pi 1_E = 1_T$, the rows should sum to 1.  And for example, $-||(0.5,0.5)||^2 = -0.5 > -1 = -||(1,0)||^2$.  So the denser vector in this case maximizes the optimization function.  In Blondel et al 2018, Fig 1 shows the regularized result as less sparse than unregularized (90% vs 94%), so that the regularized version gets better error but almost as sparse.  I am not an expert on OT, though.  This is a minor point around this particular sentence, but wondering which is correct and what I'm missing here if it's correct as written?

---

> ### Author Response · Authors · 2024-04-10
>
> Thanks for your response. You're right that the constraint $|\Pi[:, r]|_0 \leq C$ is a sparsity constraint. However, we wanted to clarify that even without this constraint, quadratic regularized optimal transport yields a sparse transportation plan. The role of the constraint $|\Pi[:, r]|_0 \leq C$ is just to ensure that the resulting solution is not only sparse (it is already sparse with quadratic regularization) but also sparse in a structured way, with no column with more than $C$ nonzeros.
>
> To explain why quadratic regularization can lead to a sparse transportation plan even without the constraint $|\Pi[:, r]|_0 \leq C$, we provide a brief answer here; for more details, see the "Squared 2-norm" section on page 4 of [Blondel et al. (2018)](https://proceedings.mlr.press/v84/blondel18a/blondel18a.pdf). Essentially, to obtain the transportation plan of the quadratically regularized OT, we recover it from the optimal dual variables. The closed form of the mapping from dual variables to primal variables is the ReLU of certain quantities related to the cost/similarity matrix. The ReLU introduces sparsity. About the example you raised, it's true that the squared 2-norm of (0.5, 0.5) is lower than that of (1, 0). However, this doesn't consider how it affects the inner product with the cost matrix (the other term in the objective function).

---

> > ### Comment · Reviewer_VVoS · 2024-04-10
> >
> > Thanks, but I'm still not sure about this example.  If the softmax is, for example, (0.7, 0.3), then including the dot product with the softmax,
> >
> > $$\Pi = (1, 0):~~~  objective = (1, 0)\cdot (0.7, 0.3) - 0.5 \cdot 1 = 0.7 - 0.5 = 0.2$$
> > and
> > $$\Pi = (0.5,0.5):~~~  objective = (0.5,0.5) \cdot (0.7, 0.3) - 0.5 \cdot 0.5 = 0.5 - 0.25 = 0.25$$
> >
> > So the objective is higher for the less-sparse choice in this case, whereas without the l2 term it would have been higher for the sparser case (0.7 vs 0.5 for dot product only).  So it seems this is regularizing towards being more dense for less-certain softmaxes?

---

> ### Comment · Reviewer_VVoS · 2024-04-10
>
> I read over the sentence that confused me again, which reads "This quadratic regularization, as shown in Blondel et al. (2018), promotes sparsity in the transportation plan when used with the non-negativity constraint"
>
> What may have confused me here, is that this reads as "the quadratic regularization term induces sparsity", whereas in Blondel et al, it seems that it's going back to the original unconstrained formulation with non-negativity (and away from entropy formulation with strict positivity, so no values can be 0 by construction due to log terms) is what produces exact sparsity.  The L2 regularization is added in to improve performance and make the objective smooth for better optimization.  So using this regularized objective enables exact sparsity, compared to the entropy objective.

---

> ### Author Response · Authors · 2024-04-11
>
> Thanks for raising this point. After re-reading the sentence you pointed out, we agree that our phrase "promotes sparsity" is misleading. The word "promote" is not accurate, because compared to the unregularized OT, the quadratic regularization doesn't introduce additional sparsity. The role of the quadratic regularization is to allow us to obtain an easier-to-solve dual while preserving sparsity (albeit less sparse than unregularized).
>
> Regarding your point on the non-negativity constraint, indeed it is important. This constraint is also present in the unregularized case. Entropic regularization makes the transportation plan strictly positive because the log acts as a barrier (due to this, the non-negativity constraint is redundant, since it is automatically satisfied).
>
> Based on your suggestion, we plan to rephrase the sentence to: "This quadratic regularization, as shown in Blondel et al. (2018), preserves the sparsity in the transportation plan and allows for an efficient numerical solver."

---

### Review · Reviewer_2yy7 · 2024-03-06

**Summary Of Contributions:**

This paper proposed a unified Mixture-of-Experts (MoE) formulation, where they examined 6 different MoE routers under an unanimous mathematical framework. This paper is the first to comprehensive study the transformer-based sparse and soft MoEs in computer vision, and provides clear derivations on the routing of MoEs. Through their experiments, the authors empirically showed that the success of language MoEs can be transferred to vision and that soft MoEs generally perform better than sparse MoEs, which offers practical insights as to how MoEs should be routed in vision tasks.

**Audience:**

Yes

**Claims And Evidence:**

Yes

**Requested Changes:**

It is better to provide the experiments on more vision tasks  (e.g., dense prediciton or visual multi-task benchmark) to provide more strong evidence to support the calims that  the success of language MoEs can be transferred to vision.

**Strengths And Weaknesses:**

**strengths:**

The paper is well-written and I am glad to read it. The mathematical derivations of this paper are clear and a unanimous mathematical framework that encompassed different routing of MoEs were proposed. The authors offered addressed weaknesses of sparse MoEs and put forward novel routing schemes to compensate for the uneven distributions of experts during training while maintaining efficiency.

**Weaknesses:**

The paper mainly evaluates different MoE variants on the few-shot transfer on ImageNet-1k, which I find to be insufficient as the paper claimed to empirically study MoEs’ performance in vision. Further experiments on other vision tasks (e.g., dense prediciton or visual multi-task benchmark) should be considered to amplify the results.

---

> ### Author Response · Authors · 2024-03-22
>
> Thank you for your review! We are glad to hear you enjoyed our paper.
>
> >The paper mainly evaluates different MoE variants on the few-shot transfer on ImageNet-1k, which I find to be insufficient as the paper claimed to empirically study MoEs’ performance in vision. Further experiments on other vision tasks (e.g., dense prediciton or visual multi-task benchmark) should be considered to amplify the results.
>
> We agree that it is beneficial to test the vision MoE on a broader range of tasks beyond classification. We used JFT300M pretraining and ImageNet few-shot experiments mainly because they align with tasks used in previous MoE research by [Riquelme et al. (2021)](https://arxiv.org/abs/2106.05974), [Allingham et al. (2023)](https://arxiv.org/abs/2110.03360), [Liu et al. (2023)](https://arxiv.org/abs/2209.15466), [Puigcerver et al. (2023)](https://arxiv.org/abs/2308.00951), as well as with the vanilla (densely-connected) vision transformer work by [Dosovitskiy et al. (2023)](https://arxiv.org/abs/2010.11929). Moreover, the results from JFT300M pretraining and ImageNet few-shot evaluations are indicative of the model's performance on other long-standing benchmarks, such as CIFAR-10/100 and Oxford-IIIT Pet, as noted by [Kolesnikov et al. (2023)](https://arxiv.org/abs/1912.11370). It will be helpful to incorporate non-classification tasks, such as the visual multi-task benchmark, but unfortunately, we currently lack the necessary infrastructure --- we do apologize for not being able to run multi-task experiments right now.

---

### Review · Reviewer_fYVm · 2024-03-10

**Summary Of Contributions:**

The paper presents an empirical study of routers in Mixture of Experts (MoE) models for computer vision tasks, introducing a unified MoE formulation that integrates various MoE layers through parametric routing tensors. This formulation enables systematic comparison and analysis of sparse and soft MoE models. The study includes experiments with six different routers on the JFT-300M dataset for pre-training and ImageNet-1k for downstream tasks, finding that soft MoEs generally outperform sparse ones within a fixed compute budget. Expert Choice routers typically surpass Token Choice routers in performance. The research highlights the significance of routers in vision MoE models and suggests that many routers developed for language modeling are also effective in vision tasks. The broader impact of this study is aimed at advancing MoE research by proposing a standardized testbed for future explorations, urging the public release of the code and benchmark results for community use.

**Audience:**

Yes

**Broader Impact Concerns:**

Accessibility of Code and Benchmark Results: The broader impact of this research could be significantly enhanced by making the code and benchmark results publicly available. Providing open access to these resources would democratize the advancements made by this work, allowing a wider range of researchers to engage in MoE research. A public, well-documented codebase would serve as a fair and standardized testbed, facilitating reproducibility and encouraging innovation. This transparency is vital for fostering a collaborative environment where the community can collectively push the boundaries of what is possible with MoE models.

**Claims And Evidence:**

Yes

**Requested Changes:**

Why are some fonts in orange (e.g. on top of page 2)?

**Strengths And Weaknesses:**

# Strengths
1. Unified Formulation of MoE: The introduction of a unified formulation for Mixture of Experts (MoE) models is a significant contribution to the field. This approach not only simplifies MoEs by providing a common framework to understand different architectures but also facilitates a more systematic study and comparison of various routing strategies. By harmonizing the underlying principles of MoE models, the authors enable researchers and practitioners to more easily identify the core components that drive performance in these complex systems, leading to more intuitive developments and enhancements.

2. Comprehensive Experiments: The exhaustive experiments conducted using the JFT300M dataset serve as a robust foundation for evaluating the performance of the proposed unified MoE model. These experiments are particularly valuable due to the scale and diversity of the JFT300M dataset, which ensures that the findings are broadly applicable. Furthermore, the transfer results on ImageNet provide critical insights into the generalizability of the model across different domains and datasets. This extensive empirical evaluation underscores the effectiveness of the proposed model and enhances the credibility of the research.

# Weaknesses
Reproducibility and Comparison with Original MoE Models: A notable concern with the proposed unified MoE model is the need for a direct comparison with the results reported in original MoE papers (where routing strategies unified in this framework that were originally proposed). It is crucial to establish that the unified model can reproduce or at least approximate the performances of the original models it seeks to encompass. Such a comparison would serve as a validation of the unified model's effectiveness and ensure that the implementation accurately reflects the advancements it claims to integrate. This step is essential for the research community to assess the model's reliability and for ensuring that the results are genuinely falsifiable, which is a cornerstone of scientific progress.

---

> ### Author Response · Authors · 2024-03-22
> **Reply from the authors**
>
> Thanks for your positive review!
>
> > A notable concern with the proposed unified MoE model is the need for a direct comparison with the results reported in original MoE papers (where routing strategies unified in this framework were originally proposed). It is crucial to establish that the unified model can reproduce or at least approximate the performances of the original models it seeks to encompass.
>
> We confirm that our results replicate previous work in vision MoE. Specifically, we use the identical experimental setup of [Riquelme et al. (2021)](https://arxiv.org/abs/2106.05974) and [Liu et al. (2023)](https://arxiv.org/abs/2209.15466), and our results are consistent to this work. The experimental setup in the original Soft MoE study [(Puigcerver et al., 2023)](https://arxiv.org/abs/2308.00951) differs slightly from ours and that of [Riquelme et al. (2021)](https://arxiv.org/abs/2106.05974) and [Liu et al. (2023)](https://arxiv.org/abs/2209.15466). For instance, the Soft MoE paper replaces the second half of dense layers with MoE layers, whereas [Riquelme et al. (2021)](https://arxiv.org/abs/2106.05974) and [Liu et al. (2023)](https://arxiv.org/abs/2209.15466) replace every second layer in vision transformers with a MoE layer. To ensure a fair comparison, our paper evaluates all MoE layers with the same settings of [Riquelme et al. (2021)](https://arxiv.org/abs/2106.05974) and [Liu et al. (2023)](https://arxiv.org/abs/2209.15466).
>
> >Why are some fonts in orange (e.g. on top of page 2)?
>
> Sorry about the confusion — we revised our manuscript based on a comment from reviewer VVoS, who posted the review earlier. The yellow text indicates the modifications made.

---

### Review · Reviewer_rgad · 2024-03-12

**Summary Of Contributions:**

This work proposes a unified formulation of Mixture-of-Experts (MoE), that incorporates several known options from the literature - Token Router, Expert Router, Soft MoE, as well as proposes a new option of router. All mentioned options are compared with each other on large-scale computer task via measuring the performance on upstream and downstream tasks. It is shown that Soft MoE consistently outperforms alternatives.

**Audience:**

Yes

**Claims And Evidence:**

Yes

**Requested Changes:**

Proposed changes and additions

* If I understood correctly, all models are evaluated with a fixed number of training steps and there are 2 options of capacity for each MoE formulation. I would suggest comparing the performance of the models at different training budgets, as it could be the case that one option may converge faster than another and the ranking may be dependent on the amount of training. In addition, would be interesting to verify, how does the performance of each approach depend on the number of experts, specifically, whether the gap between MoE formulations increases or decreases with the number of experts.
* Would be also nice to show comparison between the formulations of MoE in terms of **inference** latency and the overheads of routers.

*Minor*

I would suggest coloring each of the formulations - Token, Expert, SoftMoE with a fixed color and vary the marker shapes for Softmax, Sinkhorn on Figure [1] for the figure to be more illustrative.

**Strengths And Weaknesses:**

Strengths

* The problem studied is of significant importance to the research community and practitioners, as MoE formulation allows one to have a model that is both expressive and efficient to inference. This work investigates various design options of MoE and compares them in a unified setup, reliably showing superiority of one approach over another.
* Dispatch-combine tensor formulation is novel and incorporates the known variations of MoE layers as well as allows for a design of new one.
* The experimental study is quite extensive, involving models of different size and hyperparameter choices.

Weaknesses
* The comparison between token, expert and soft MoE is not novel. In the referenced work [1] Figure 3 contains detailed comparison between different expert families at different training budgets. The contribution of this work is more fine-grained-comparison involving the algorithm of router assignment, that is important, but somehow incremental. The choice of router softmax/sinkhorn seems to matter much less compared to the choice of Token/Expert/Soft MoE.
* The experimental evaluation involves only JFT upstream accuracy and ImageNet-1k few-shots. I would suggest adding full-model finetuning on ImageNet-1k and evaluation on a couple more datasets from VTAB benchmark.

---
[1] Puigcerver, Joan, et al. "From sparse to soft mixtures of experts." arXiv preprint arXiv:2308.00951 (2023).

[2] https://github.com/google-research/task_adaptation

---

> ### Author Response · Authors · 2024-03-22
> **Reply from the authors**
>
> Thank you for your review!
>
> > The contribution of this work is more fine-grained-comparison involving the algorithm of router assignment, that is important, but somehow incremental.
>
> Indeed, in this paper we offer a more detailed comparison of MoE routers compared to [Puigcerver et al., 2023](https://arxiv.org/abs/2308.00951). Specifically, we decouple the influence of **(i)** the token-expert affinity computation step (Softmax vs. Sinkhorn vs. sparsity-constrained OT) and **(ii)** the router tensor allocation step (Token Choice vs. Expert Choice vs. SoftMoE). These new studies show that the design choices for token-expert affinity computation in (i) are not the primary factor for performance improvement. However, the choice of router tensor allocation methods in (ii) significantly impacts performance.
>
> > I would suggest adding full-model finetuning on ImageNet-1k and evaluation on a couple more datasets from VTAB benchmark.
>
> Thank you for these concrete suggestions for experiments. Currently, we have the finetuning results of some routers but not all of them. The performance of finetuning is usually highly correlated with few shot results, which we present in our paper. We’ll check if we have the capacity for further experimentation.
>
> > I would suggest comparing the performance of the models at different training budgets, as it could be the case that one option may converge faster than another and the ranking may be dependent on the amount of training. In addition, would be interesting to verify, how does the performance of each approach depend on the number of experts, specifically, whether the gap between MoE formulations increases or decreases with the number of experts.
>
> Yes, in the present paper, we assess the impact of the routers while keeping other configurations (training budget per router and the number of experts used per router) constant. A sweep of these configurations requires combinatorially more runs and evaluations, which is quite resource-intensive. This could be highly informative, but a detailed exploration might fit better in a future paper on vision MoE scaling laws, similar to the scaling-law paper for language MoE by [Clark et al., (2022)](https://proceedings.mlr.press/v162/clark22a/clark22a.pdf).
>
> > Would be also nice to show comparison between the formulations of MoE in terms of inference latency and the overheads of routers.
>
> This is a good point and we’ll add a remark about this in the latest version of the manuscript. Indeed, the compute-accuracy tradeoff depicted in Figures 1 and 2 measures the total training time (in TPUv3-days) for VMoE with various routers, including both forward and backward pass times. The relative ranking of the routers would remain consistent if we reported inference time instead. This is because routers that are visibly slower than others use iterative optimal transport algorithms in the forward pass. Since the optimal transport algorithm is skipped in the backward pass (as discussed in Section 2.2 and further explored in Section 3.4), their backward pass is comparatively fast and does not contribute to differentiating the overall training time.

---

### Decision · Action_Editor_nBcq · 2024-04-17

**Recommendation:** Accept as is

**Comment:**

This paper proposes a systematic framework for MOE, which includes various strategies to construct a powerful MOE model. All the reviewers approve of the manuscript's contribution and its clear presentation. During the rebuttal phase, the authors also addressed the reviewers' queries regarding experimental details and other aspects, supporting the acceptance of the paper.

Overall, this manuscript fits the TMLR acceptance criteria and I recommend accepting this paper.

**Audience:**

Yes.  Individuals in TMLR's audience will be interested in this paper.

**Claims And Evidence:**

Yes. The claims are well supported by extensive experiments and analysis.